# Theory of correlated insulating behaviour and spin-triplet superconductivity in twisted double bilayer graphene

Jong Yeon Lee[1,2], Eslam Khalaf[1,2], Shang Liu[1], Xiaomeng Liu[1], Zeyu Hao[1], Philip Kim[1] & Ashvin Vishwanath[1]*

Two graphene monolayers twisted by a small magic angle exhibit nearly flat bands, leading to correlated electronic states. Here we study a related but different system with reduced symmetry - twisted double bilayer graphene (TDBG), consisting of two Bernal stacked bilayer graphenes, twisted with respect to one another. Unlike the monolayer case, we show that isolated flat bands only appear on application of a vertical displacement field. We construct a phase diagram as a function of twist angle and displacement field, incorporating interactions via a Hartree-Fock approximation. At half-filling, ferromagnetic insulators are stabilized with valley Chern number $C_v = \pm 2$. Upon doping, ferromagnetic fluctuations are argued to lead to spin-triplet superconductivity from pairing between opposite valleys. We highlight a novel orbital effect arising from in-plane fields plays an important role in interpreting experiments. Combined with recent experimental findings, our results establish TDBG as a tunable platform to realize rare phases in conventional solids.

[1] Department of Physics, Harvard University, Cambridge, MA 02138, USA. [2]These authors contributed equally: Jong Yeon Lee, Eslam Khalaf.
*email: avishwamath@g.harvard.edu

The recent discovery of correlated insulating states and superconductivity in twisted bilayer graphene (TBG)[1–4] has opened a new window to exploring strong correlation effects in systems whose doping can be easily tuned, enabling the exploration of a rich range of interaction-driven phenomena. Although the underlying reason for the correlated physics is understood to arise from a relatively narrow electronic bandwidth induced by the long wavelength Moiré pattern[5,6], several details, including the symmetry breaking within the insulating phase and the nature and mechanism of pairing in the neighboring superconductor, remain under debate[7–19]. One of the difficulties in addressing these questions arises from the complexity of the theoretical treatment of TBG, which involves at least a pair of narrow bands per spin per valley with a symmetry-protected band touching, leading to eight bands in total. On top of that, the limited tunability of the band structure makes it experimentally difficult to explore the dependence of different phases on microscopic parameters.

Motivated by recent experimental report[20], we study a related system—twisted double-bilayer graphene (TDBG)—which consists of a pair of bilayer-graphene sheets, twisted with respect to one another with AB–AB-stacking structure. Due to the absence of $C_2$ rotation symmetry, TDBG has a lower symmetry compared with TBG, which simplifies the problem by removing the band touching at the Dirac points, leading to a low energy effective description involving one rather than two narrow bands per spin and valley. Moreover, the band separation can be controlled by applying a vertical displacement field enabling the exploration of different regimes of band isolation and bandwidth within the same device.

We identify three main ingredients necessary to explain the emergence of insulating and superconducting behavior in TDBG. First, we perform an accurate calculation of the single-particle band structure to identify ranges of displacement field and twist angle for which a single band is isolated and relatively flat. We show that lattice relaxation, known to be important in TBG[21,22], as well as several other effects such as trigonal warping, which are absent in TBG, significantly influence the band structure in TDBG, in excellent agreement with experiments. Moreover, we identify a hitherto-neglected in-plane orbital effect which is used to explain the experimentally observed deviation of the in-plane $g$ factor from 2[20], as well as the effect of in-plane field on superconducting $T_c$.

Second, we address the key question of the nature of the interaction-driven insulating state. The similarity between the phase diagram of TBG to that of cuprates was invoked to argue that Mott physics is the underlying mechanism responsible for the correlated insulator[1,7,12]. On the other hand, a different route to correlated insulators is observed in graphene quantum-Hall systems, for instance, when the spin and valley degeneracy of the Landau levels are spontaneously broken by interactions[23]. This usually leads to ferromagnetic insulators, which are otherwise rare in correlated solids where antiferromagnetic order is the norm. For similar reasons, in the TDBG with nonzero valley Chern number, ferromagnetism may be preferred[24] at integer fillings. The situation here is reminiscent of strained graphene, where a suitably chosen strain profile leads to Landau levels arising from the opposite strain magnetic fields applied on the two valleys[25]. At partial fillings that are integers, ferromagnetic ground states were obtained with repulsive interactions[26], and we show that a similar scenario is likely to occur here in TDBG. Indeed a related ground state with spontaneous quantum-Hall response, although metallic, was observed in the twisted monolayer-monolayer graphene (TBG) with $C_2$-breaking substrate potentials[13,19,24,27–29].

Third, we investigate the nature of the superconducting phase by highlighting that the valley degree of freedom, which behaves

as a pseudospin, allows for exotic pairing possibilities which are relatively rare in other materials. In particular, we show that spin triplet with valley-singlet pairing, which is momentum-independent within each valley, is favored. We investigate the consequences of such scenario and show it can be used to explain the measured dependence of $T_c$ on in-plane field[20].

## Results

**Single-particle physics.** We consider a system consisting of two AB-stacked graphene bilayers twisted relative to AB–AB stacking by a small angle $\theta$, illustrated in Fig. 1. For a detailed discussion on the Hamiltonian and model parameters, see the Methods section. The bottom layer of the top BLG and the top layer of the bottom BLG are coupled via Moiré hopping between $AA$ and $AB$ sites, parametrized by $(w_0, w_1)$[21,22]. In the original Bistritzer–Macdonald model, $w_0$ and $w_1$ are taken to be equal[30]. However, in a realistic twisted model, the ratio $r \equiv w_0/w_1$ is smaller than one due to the lattice relaxation which expands (shrinks) AB (AA) regions. In TBG, $r$ is taken to be ~0.75 for the first magic angle[21,22]. Here, we similarly include lattice relaxations by taking $r$ to be <1. This is crucial for the existence of a gap between first and second conduction (valence) bands in TDBG which is necessary to explain the band insulator at $\nu = \pm 4$ filling. In this work, we take $(w_0, w_1) = (88, 100)$ meV corresponding to $r = 0.88$. For different values of $(w_0, w_1)$, we obtained qualitatively similar features (Methods).

Unlike TBG, a realistic description of TDBG does not exhibit magic-angle physics whose origin is the vicinity to a chiral symmetric model with perfectly flat bands at specific angles[31,32]. In the quadratic approximation of the bilayer-graphene dispersion, the first conduction and valence bands in TDBG become almost perfectly flat at the angle $\theta \approx 1.05$[24]. However, once trigonal warping ($\gamma_3$) and particle–hole asymmetry ($\gamma_4$) terms are included, the flat bands acquire a significant dispersion and become overlapped with each other (Fig. 2a, b). Theses bands can only be separated by applying a strong enough gate voltage between top and bottom layers (Fig. 2c). Using numerical simulations, we identify the parameter space of twist angle $\theta$ and applied voltage $U$ where the first conduction band is isolated (Fig. 3a). On the other hand, we find that there is barely any regime where the first valence band is isolated (Fig. 3c). Such a particle–hole asymmetry in the band structure is originated from $\gamma_4$ and $\Delta$ terms. The results are consistent with the experimental findings[20], showing that the system at charge neutrality remains metallic unless a rather large vertical electric field is applied. Furthermore, a correlated insulating phase is only observed on

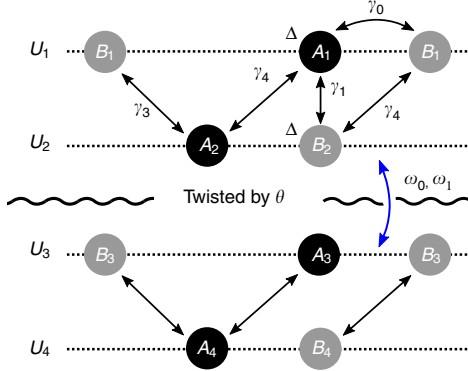

**Fig. 1** Twisted double BLG model (AB–AB stacking) with the gating voltage $U$ across the system. Throughout this work, we assume the voltage drop across the layers is uniform, $U_i - U_{i+1} = U/3$.

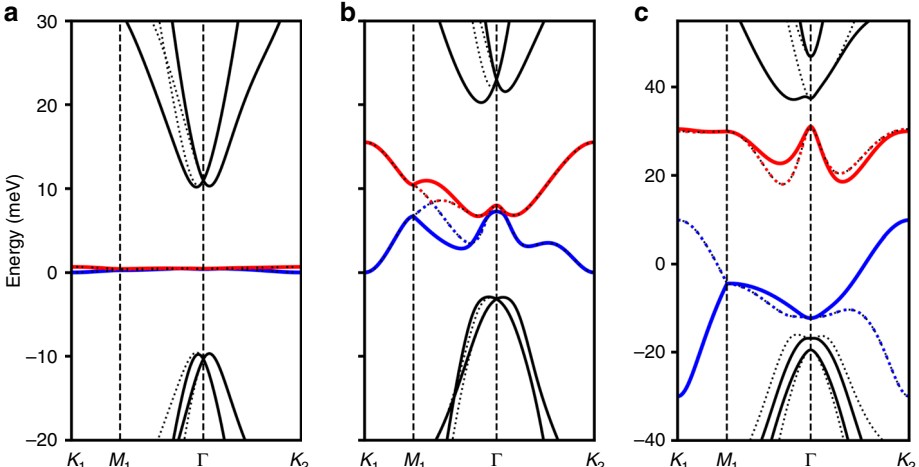

**Fig. 2** Moiré band structures of TDBG. **a**, **b** At $(\theta, U) = (1.05°, 0)$. Solid (dotted) line represents the band originated from $\mathbf{K}_+$ ($\mathbf{K}_-$) valley. Red, blue and black represent conduction, valence, and the other bands, respectively. **a** The band structure for the idealized model with only $\gamma_0$ and $\gamma_1$ being nonzero. The flat band is observed with the bandwidth 0.25 meV. **b** The band structure for the realistic model with overlapping bands. The "magic angle" does not exist in this case. **c** Moiré band structure at $(\theta, U) = (1.33°, 60\,\text{meV})$. The first conduction band (red) is isolated and relatively flat.

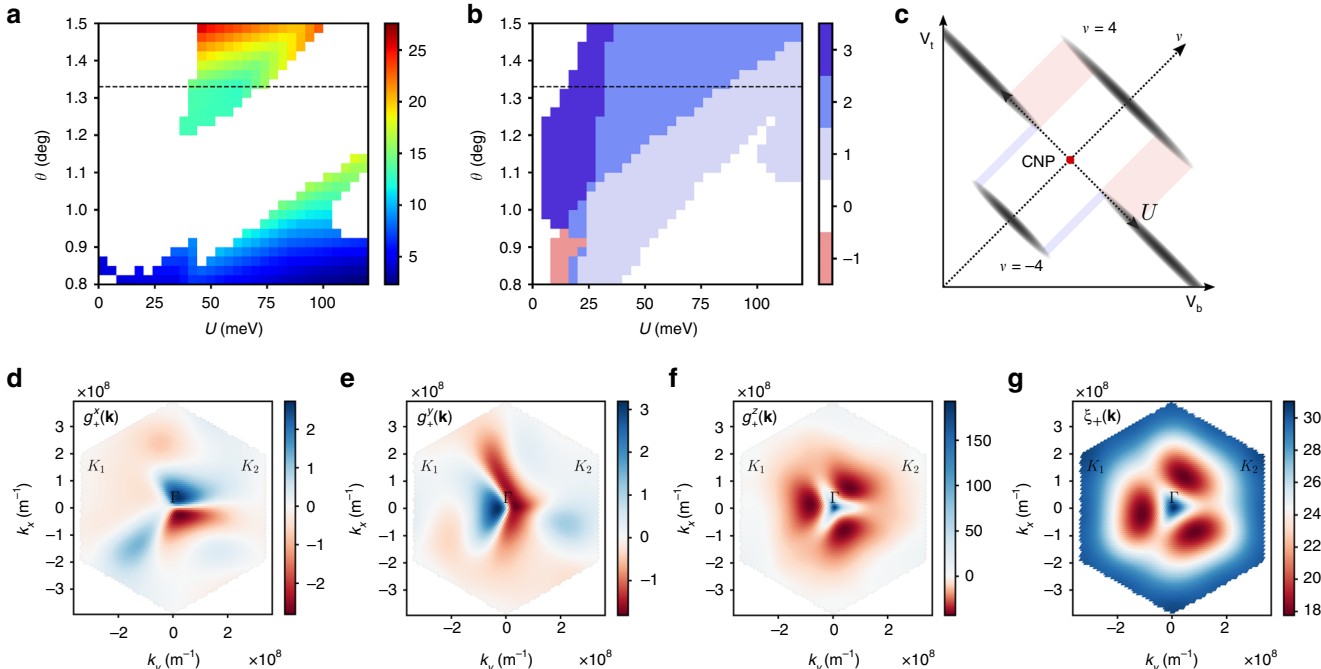

**Fig. 3** Summary of single-particle calculations of TDBG. **a** Isolation region for the first conduction band (colored) with the bandwith indicated by the color. We observe two seperate isolation regions for $\theta$ smaller or larger than 1.1°. The former is not very robust, and is sensitive to fine-tuning of parameters whereas the latter is very robust and is associated with a valley Chern number of 2 (See **b**). **b** The Chern number of the first conduction band from $\mathbf{K}_+$ valley. Note, the Chern number is defined as long as a direct bandgap is present. **c** A schematic plot for the insulating (black) regions and the first conduction/valence band isolated region (red/blue) in the TDBG at $\theta = 1.33°$. The red dot is charge neutrality point (CNP). In the shaded region, strongly correlated physics is expected near integer fillings. Asymmetry between electron and hole dopings is predicted from the theory. **d**–**g** Color plots for $g$-factor associated with orbital magnetic effects $g_+^x(\mathbf{k})$, $g_+^y(\mathbf{k})$, $g_+^z(\mathbf{k})$, and single-particle dispersion $\xi_+(\mathbf{k})$ over the Moiré Brillouin zone for the first conduction band at $(\theta, U) = (1.33°, 60\,\text{meV})$, where the band is isolated. $g^{x,y,z}(\mathbf{k})$ are in the unit of $\mu_B$, and $\xi(\mathbf{k})$ is in the unit of meV. Both $g^x$ and $g^y$ vanish at high symmetric points $\Gamma$, $K_1$, and $K_2$.

electron-doping side, consistent with the theoretically expected particle–hole asymmetry. Note that the bandwidth is not as flat as that of magic-angle TBG. However, the bandwidth is still small compared with the interaction scale which implies that strongly correlated physics can still arise. Indeed, there is some debate regarding the bandwidth of magic-angle TBG itself, with reported bandwidths ranging from 10 to 40 meV[33].

Another crucial difference compared with TBG is the absence of twofold rotational symmetry, which protects the Dirac points in TBG. As a result, the physics of TDBG is controlled by a single narrow band (per spin per valley) rather than two as in TBG. The TDBG Hamiltonian has the following symmetries (i) threefold rotation symmetry $C_3$, (ii) time-reversal symmetry $\mathcal{T}$, and, (iii) mirror reflection about the $x$-axis $M_y$, which only exists in the

absence of vertical electric field, and (iv) SU(2) spin-rotation symmetry. Finally, we assume that in the small angle limit, there is valley-charge-conservation symmetry $U(1)_v$, arising from the decoupling of Moiré and atomic lattice-scale physics.

In addition, the conduction band within each valley carries a nonzero Chern number. In ordinary condensed matter systems, $\mathcal{T}$-symmetry forbids the existence of Chern bands. However, in Moiré systems, Chern bands carrying opposite Chern numbers for opposite valleys can arise due to the valley decoupling. The overall system still satisfies $\mathcal{T}$-symmetry, which exchanges the two valleys. Therefore, spontaneous valley polarization would lead to a Chern band without explicitly breaking $\mathcal{T}$-symmetry[13,24,26,29]. At $U = 0$, the reflection symmetry $M_y$ enforces $C = 0$ for both valleys. At $U \neq 0$, the conduction band develops a nonvanishing Chern number computed numerically in Fig. 3c which is equal to $\pm 2$ for the parameter region corresponding to band isolation. The evolution of Chern number as a function of $U$ is further confirmed using symmetry indicator (Methods). This can be also understood from the well-known behavior of a AB-stacked bilayer graphene under an electric field. Under the electric field, the bilayer graphene becomes gapped and accumulates opposite Berry curvatures at $\mathbf{K}_+$ and $\mathbf{K}_-$ valleys, which amounts to a Chern number $C_v = \pm 2$ for each valley[34–37].

Finally, we discuss the effect of applied magnetic field which influences the single-particle physics in two distinct ways. First, it couples to the electron spin via Zeeman effect leading to the splitting of bands with opposite spin by $2\mu_B B$. Second, it couples to the electron orbital motion leading to modifications in the band structure. For out-of-plane field, the orbital effect arises from the magnetic field coupling to the planar motion of the electron[38,39]. It leads to an energy correction of $\mu_B g^z_\tau(\mathbf{k}) B_z$, with a $\mathbf{k}$-dependent $g$-factor $g^z_\tau(\mathbf{k})$ satisfying $g^z_{-\tau}(-\mathbf{k}) = -g^z_\tau(\mathbf{k})$ due to time-reversal symmetry ($\tau$ is a valley index). As shown in Fig. 3f, $g^z_\tau(\mathbf{k})$ can be much larger than the Zeeman effect. For in-plane field, the orbital effect arises from coupling to the interlayer motion of electrons. For an in-plane field $\mathbf{B}$, we can choose the gauge $\mathbf{A}(z) = -z \times \mathbf{B}$ which does not depend on $x$ or $y$, thus preserving the Moiré translation symmetry. The resulting change in the hopping parameters is obtained by the Peierl's substitution, effectively providing an additional momentum shift of $-\frac{e}{\hbar} \frac{(l+m)d}{2} \, e_z \times \mathbf{B}$ to the hopping connecting layers from $l$ to $m$, where $d$ is the interlayer separation (see the Methods section). This leads to an energy correction of the form $\mu_B(g^x_\tau(\mathbf{k}) B_x + g^y_\tau(\mathbf{k}) B_y)$ to the leading order in $\mathbf{B}$ with $g^{x,y}_{-\tau}(-\mathbf{k}) = -g^{x,y}_\tau(\mathbf{k})$. The orbital effect due to in-plane field amounts to a very small relative momentum shift $\sim \frac{eda}{\hbar} \approx 10^{-5}$. However, it cannot be neglected since it is of the same order of magnitude as the Zeeman effect, $\frac{e v_F d}{\mu_B} \sim 1$ (see Fig. 3d, e). In general, the in-plane orbital contribution changes the band dispersion due to its $\mathbf{k}$-dependence, whereas the Zeeman effect shifts the entire band uniformly. Moreover, it acts oppositely for different valleys. These properties can be crucial in understanding the effect of in-plane field on the insulating gap and the superconducting temperature (see the Methods section and Supplementary Note 6).

## Correlated insulating states

In the band isolation regime, the first conduction band carries a nonzero Chern number as shown in Fig. 3a, b which prevents the existence of exponentially localized Wannier functions[40]. As a result, one cannot construct a Hubbard model for the band unless valley symmetry is broken, or the model is enlarged to include more bands so that the net Chern number is zero. Instead of seeking a complicated real-space description, we discuss the interaction effects in the

momentum space, as in the case of quantum-Hall ferromagnetism. One major consequence of the absence of localized Wannier orbitals is the inadequacy of the Mott picture, where the insulating phase is driven by strong repulsion between localized orbitals. Thus, we will use the terminology, correlated insulator to refer to the interaction-driven insulating phase for the following physics.

In order to uncover the nature of the possible correlated insulating states at half and quarter-filling[20], we perform a self-consistent Hartree–Fock mean-field theory similar to the one employed in ref. [8,24]. Below, we sketch the derivation from the microscopic theory, relegating most details to Supplementary Notes 2 and 3. The interacting Hamiltonian in momentum space is given by

$$\mathcal{H}_{\text{int}} = \frac{1}{2 \, \text{Vol}} \sum_q \hat{\rho}(q) V(q) \hat{\rho}(-q), \qquad (1)$$

where $V(q)$ is the Fourier-transformed screened Coulomb interaction[41,42]. Since the screening coming from the distance between the system and the gate is comparable with the Moiré length scale, the screening length can be important for the interaction effects. The density $\hat{\rho}(q)$ consists of an intravalley part $\rho^+ \sim c^\dagger_\pm c_\pm$ and an intervalley part $\rho^- \sim c^\dagger_\pm c_\mp$, where $c^\dagger_\pm$ is the electron creation operator for $\mathbf{K}_\pm$ valley. The latter contribution arises from the small coupling between opposite valleys and gives rise to an intervalley Hund's coupling term.

The resulting Hamiltonian consists of two parts, $\mathcal{H}_{\text{int}} = \mathcal{H}_0 + \mathcal{H}_J$, where $\mathcal{H}_0$ contains the coupling between intravalley densities $\rho^+ \rho^+$, whereas $\mathcal{H}_J$ contains the coupling between intervalley densities $\rho^- \rho^-$. Rough estimation for the relative energy scales for $H_0$ and $H_J$ gives $V_0 \sim 35$ meV and $J \sim 0.6$ meV for the experimentally relevant regime. Although $H_J$ is significantly smaller than $H_0$, it breaks the symmetry of the model down from two independent SU(2) spin-rotation symmetries for each valley to a single SU(2). Thus, it can lift the degeneracy between some symmetry breaking states which are degenerate on the level of the $H_0$. Indeed, we found that $H_J$ favors the spin alignment between opposite valleys and can be written in the form of intervalley Hund's coupling as in ref. [24].

Within the self-consistent Hartree–Fock mean-field theory, we consider the order parameter defined as

$$\langle c^\dagger_{\sigma,\tau}(\mathbf{k}) c_{\sigma',\tau'}(\mathbf{k}') \rangle = M_{\sigma\tau,\sigma'\tau'}(\mathbf{k}) \delta_{\mathbf{k},\mathbf{k}'}. \qquad (2)$$

For a gapped phase, matrix $M(\mathbf{k})$ must be a projector, i.e., $M(\mathbf{k})^2 = M(\mathbf{k})$ satisfying $\text{tr} \, M(\mathbf{k}) = \nu$ for all $\mathbf{k}$. Given that there are four flavors of fermions due to spin ($\sigma$) and valley ($\tau$) degeneracies, any possible order parameter $M$ can be expanded in terms of the generators of SU(4) $\sigma_i \otimes \tau_j$, which can be grouped based on their symmetry breaking into five categories: (i) $\{\sigma_0 \tau_z\}$ only breaks $\mathcal{T}$ and corresponds to a valley-polarized (VP) state, (ii) $\{\sigma_{x,y,z} \tau_0\}$ breaks spin-rotation symmetry and correspond to a spin-polarized (SP) state. (iii) $\{\sigma_{x,y,z} \tau_z\}$ breaks both spin rotation and time-reversal (but preserve some combination of the two) and corresponds to a spin-valley locked (SVL) state, (iv) $\{\sigma_0 \tau_{x,y}\}$ breaks $U(1)$ valley-charge conservation and corresponds to an intervalley coherent (IVC) state, and (v) $\{\sigma_{x,y,z} \tau_{x,y}\}$ breaks both spin rotation and U(1)$_v$ valley-charge conservation, corresponds to spin-IVC locked (SIVCL) state (see Table 1). We note that any of these orders may break or preserve $C_3$ symmetry depending on its $\mathbf{k}$ dependence.

The results of the self-consistent Hartree–Fock calculation are summarized in the following (Supplementary Note 3). Restricting ourselves to translation-symmetric gapped states, we find there are five options: SP, VP, SVL, IVC, and SIVCL at half-filling

**Table 1 Symmetry broken states and the remaining symmetries for all possible translation-symmetric gapped states at $\nu = 1$, 2, 3.**

| $\nu = 2$ | Example of the state | Symmetry |
|---|---|---|
| SP | $\lvert \uparrow \mathbf{K}_+ \rangle \otimes \lvert \uparrow \mathbf{K}_- \rangle$ | $U(1)_z$, $U(1)_v$, $\mathcal{T}$ |
| VP | $\lvert \uparrow \mathbf{K}_+ \rangle \otimes \lvert \downarrow \mathbf{K}_+ \rangle$ | $SU(2)$, $\mathcal{T}$ |
| SVL | $\lvert \uparrow \mathbf{K}_+ \rangle \otimes \lvert \downarrow \mathbf{K}_- \rangle$ | $U(1)_z$, $U(1)_v$, $\mathcal{T}'$ |
| IVC | $(\lvert \uparrow \mathbf{K}_+ \rangle + e^{i\theta} \lvert \uparrow \mathbf{K}_- \rangle) \otimes (\lvert \downarrow \mathbf{K}_+ \rangle + e^{i\theta} \lvert \downarrow \mathbf{K}_- \rangle)$ | $SU(2)$, $\mathcal{T}$ |
| SIVCL | $(\lvert \uparrow \mathbf{K}_+ \rangle + e^{i\theta} \lvert \downarrow \mathbf{K}_- \rangle) \otimes (\lvert \downarrow \mathbf{K}_+ \rangle + e^{i\theta} \lvert \uparrow \mathbf{K}_- \rangle)$ | $U(1)_z$, $Z_2^{xv}$, $\mathcal{T}$ |

| $\nu = 1, 3$ | Example of the state | Symmetry |
|---|---|---|
| SVP | $\lvert \uparrow \mathbf{K}_+ \rangle$ | $U(1)_z$, $U(1)_v$ |
| SPIVC | $\lvert \uparrow \mathbf{K}_+ \rangle + e^{i\theta} \lvert \uparrow \mathbf{K}_- \rangle$ | $U(1)_z$, $\mathcal{T}$ |
| SVLIVC | $\lvert \uparrow \mathbf{K}_+ \rangle + e^{i\theta} \lvert \downarrow \mathbf{K}_- \rangle$ | $Z_2^{zv}$, $\mathcal{T}'$ |

The similar table with the form of $M(\mathbf{k})$ and symmetry generators is in the Supplementary Table 1. Here, $\mathcal{T}$ is the spinless time-reversal $\mathcal{T} = \tau_x \mathcal{K}$ squaring to $+1$ whereas $\mathcal{T}'$ is the spinful time-reversal $\mathcal{T}' = i\sigma_y \mathcal{T}$ squaring to $-1$ (with $\mathcal{K}$ denoting complex conjugation). $U(1)_{x,y,z}^{\theta} = e^{i\theta\sigma_{x,y,z}/2}$ denotes spin rotation around the $x$, $y$, $z$ axis by an angle $\theta$ whereas $U(1)_v^{\theta} = e^{i\theta\tau_z/2}$ denotes rotation in the valley $x - y$ plane by an angle $\theta$. Finally, $Z_2^{z,v}$ is generated by the combined rotation $U(1)_z^{\pi} U(1)_v^{\pi}$.

$\nu = 2$ and three options: spin-valley-polarized (SVP), spin-polarized-IVC (SPIVC), and spin-valley-locked-IVC (SVLIVC) at quarter-filling $\nu = 1, 3$, as in Table 1. By solving the Hartree–Fock self-consistency condition, the ground-state energy $E$ and the correlation gap $\Delta$ are computed for different states (Fig. 4a). Let us first consider what happens in the absence of intervalley Hund's coupling. In this case, we find that the SP and SVL states at half-filling and similarly the SPIVC and SVLIVC states at quarter-filling are exactly degenerate since they are related by a spin rotation in one of the valleys. Similarly, due to the enlarged symmetry of the mean-field Hamiltonian, the SP and VP states and the IVC and SIVCL states have the same energy. Thus, we only need to numerically investigate the competition between SP and IVC at half-filling and SVP and SPIVC at quarter-filling. The result of such numerical investigation is shown in Fig. 4a, where we clearly see that SP has a lower energy than that of the IVC in most of the parameter regime. Similar results apply for the competition between SVP and SPIVC at quarter-filling. The correlation-induced gap $\Delta$ for the SP state in the band isolation region ranges between 4 and 8 meV (see Fig. 4b).

To understand the reason why IVC order is energetically unfavorable, we can employ the argument of ref. [29] as follows. IVC order between two valleys with opposite Chern number $C$ is equivalent after a particle–hole transformation in one of the valleys to superconducting pairing between bands with the same Chern number i.e., a superconductor in a background magnetic field. This means that the order parameter necessarily includes $\lvert C \rvert$ vortices within the Brillouin zone leading to increased energy. A more detailed analytic treatment of the energy competition between SP and IVC is provided in the Supplementary Note 4.

The inclusion of the effect of intervalley Hund's coupling alters the competition between the phases as follows. First, since the term is ferromagnetic, it lowers the energy of the SP state, favoring the SP state over the VP-state, which is in turn favored over the SVL-state. Second, it lowers the energy of the filled bands for the SP state at half-filling, thus increasing $\Delta_{SP}$. On the other hand, it reduces the energy of some of the empty bands for the VP-state, reducing $\Delta_{VP}$ (see Fig. 4c, e). The Hund's coupling term similarly reduces $\Delta_{SVP}$ at quarter-filling by lowering the energy of

one of the excited states (see Fig. 4d). We note here that the reduction of the correlated gap at quarter-filling relative to that at half-filling may explain why the former is more difficult to observe experimentally compared with the latter and requires the application of a magnetic field[20].

In the presence of an in-plane field, the gap of the SP-phase at half-filling is expected to grow with a slope consistent with the Zeeman $g = 2$ factor. However, the orbital effect discussed earlier leads to a reduction in the effective $g$-factor by 20–50% depending on the band structure details (Fig. 4e), which is in agreement with the experimental data[20]. From the numerical calculation, we confirmed that such a reduction in gap also depends on the in-plane field direction, which exhibits threefold periodicity (see the Methods section). Therefore, the orbital effect can be directly verified in a rotating in-plane field setup, where we predict the modulation of the $g$-factor with period $2\pi/3$ in the angle.

**Superconductivity**. When the correlated insulator is doped away from half-filling, a superconducting phase is observed below 3.5 K[20]. Our proposed scenario for the observed superconductivity is illustrated in Fig. 5a, where pairing takes place between time-reversal partners in opposite valley. Such an intervalley pairing between time-reversal partners has also been proposed[43–45] and observed in transition metal dichalcogenides (TMD)[46]. However, unlike in TMD, where strong spin–orbit coupling implies a locking between spin and valley, here the proposed pairing takes place between the electrons with the same spin. To understand this, we first note that doping a spin-polarized insulator is expected to give rise to a ferromagnetic metal with spin-split Fermi surface. Similar to other ferromagnetic metals[47–49], ferromagnetic spin fluctuations can act as a pairing glue responsible for superconductivity[50]. This motivates the following simplified Hamiltonian,

$$\mathcal{H} = \sum_{\mathbf{k},\tau,\sigma} c_{\sigma,\tau,\mathbf{k}}^{\dagger} \xi_{\sigma,\tau,\mathbf{k}} c_{\sigma,\tau,\mathbf{k}} - g \sum_q S_q \cdot S_{-q}, \tag{3}$$

where the spin operator $S_q^a = \sum_{\mathbf{k},\tau,\sigma,\sigma'} c_{\sigma,\tau,\mathbf{k}+q}^{\dagger} \sigma_{\sigma,\sigma'}^a c_{\sigma',\tau,\mathbf{k}}$. This Hamiltonian can be obtained within an RPA treatment by identifying the ferromagnetic order as the leading instability in the doped itinerant phase. The ferromagnetic susceptibility is peaked at $q = 0$, which justifies a $\mathbf{k}$-independent coupling.

Next, we consider the simplest possible intervalley superconducting pairing function $\Delta$, which is $\mathbf{k}$-independent ($s$-wave) within each valley. Note, however, that the overall orbital symmetry incorporating both momentum and valley may still be anti-symmetric, e.g., $p$-wave. For the proposed pairing, $\Delta$ is proportional to $\tau_x$ or $\tau_y$ corresponding to valley triplet or singlet, respectively. The overall antisymmetry of $\Delta$ implies that the former scenario corresponds to a spin-singlet $i\sigma_y$, whereas the latter corresponds to a spin triplet $i\sigma_y d \cdot \boldsymbol{\sigma}$. Here, $d$ is the vector which captures the direction of the spin state. To see which of these is the dominant pairing channel, it is useful to decouple the interaction in the pairing channel as

$$\mathcal{H}_{\text{int}} = -g \sum_{\mathbf{k},q} \text{tr} \, (\boldsymbol{\sigma} \Delta_{\mathbf{k}}) \cdot (\boldsymbol{\sigma}^T \Delta_{\mathbf{k}+q}^{\dagger}) \tag{4}$$

We now assume $\mathbf{k}$-independent $\Delta$ and decompose it into spin-singlet/velly triplet $\Delta_s$ and spin triplet/valley-singlet $\Delta_t$. We now use

$$\boldsymbol{\sigma} \cdot (\Delta_{t,s} \boldsymbol{\sigma}^T) = \lambda_{t,s} \Delta_{t,s}, \tag{5}$$

where $\lambda_t = 1$ and $\lambda_s = -3$. This means that the interaction is repulsive in the singlet channel, and attractive in the triplet channel making the latter the dominant pairing channel. A more

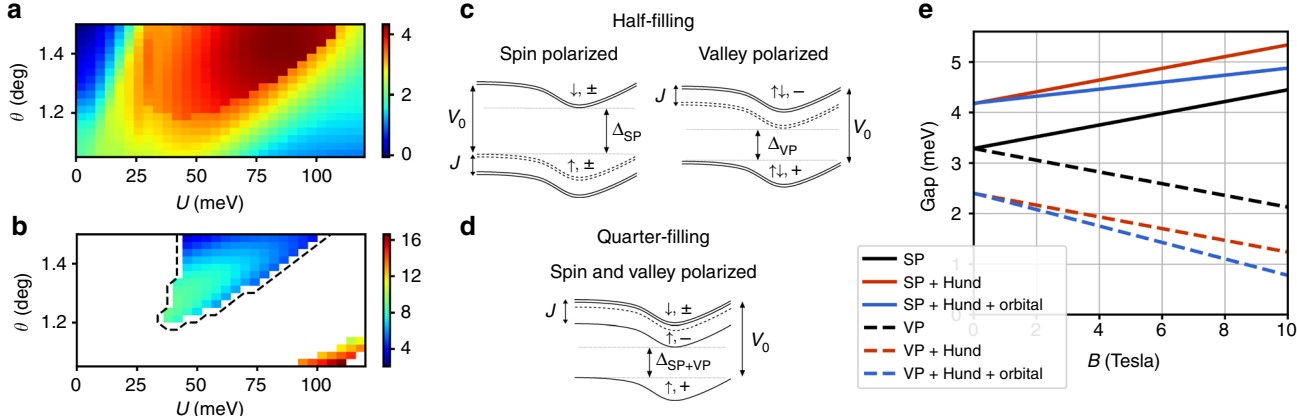

**Fig. 4** The results of the Hartree–Fock calculation. **a** Color plot (meV) for $E_{IVC} - E_{VP}$ per electron. **b** Color plot of self-consistency gap $\Delta_{SP/VP}$ for the SP/VP-state in the band isolated region. (No $J$-term included) **c**, **d** Effect of the intervalley Hund's coupling ($J$-term) on the gap for spin- and valley-polarized phases at half and quater fillings, respectively. At half-filling, $J$-term increases (decreases) $\Delta_{SP}$ ($\Delta_{VP}$). At quarter-filling, $J$-term reduces the gap to the next-excited state, making the quarter-filled insulator (SP + VP) less stable than the half-filled (SP) one. **e** The correlated gap $\Delta$ for half-filling insulators (SP, VP) as a function of in-plane $B_x$-field. $(\theta, U) = (1.33°, 60\,\text{meV})$. Solid lines for SP state and dotted lines for VP-state. Zeeman effect would increase (decrease) $\Delta$ for the SP (VP) state with increasing $B$. The valley orbital effect $g^{x,y}(\mathbf{k})$ leads to a linear decrease in the gap with field, thus effectively decreasing (increasing) the $g$-factor for the SPS (VP) state.

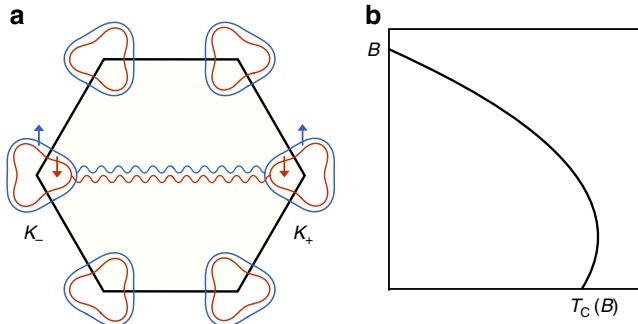

**Fig. 5** Spin-triplet superconductivity. **a** Triplet paring between opposite valleys, $c_{\sigma,+}(\mathbf{k})$ and $c_{\sigma,-}(-\mathbf{k})$ with exact energy match. **b** Schematic plot for the $T_c$ as a function of $B$-field.

detailed discussion of these pairing channels within the linearized BCS equation is provided in the Supplementary Note 5.

We highlight here that spin-triplet pairing is only known to occur in liquid He₃[51] and a few Uranium compounds[47–49], as it requires pairing that varies over the Fermi surface (eg. $p$-wave) which is likely to be energetically unfavorable in typical solids. The existence of the valley degree of freedom here enables us to evade this difficulty and obtain a spin-triplet valley-singlet order parameter even for a $\mathbf{k}$-independent interaction.

The experimental consequences of the proposed spin-triplet valley-singlet superconductivity can be investigated by writing the Ginzburg–Landau free energy for the order parameter $\Delta = \tau_y \sigma_y d \cdot \boldsymbol{\sigma}$ in the presence of a magnetic field $\mathbf{B}$. Restricting ourselves to terms up to quartic order in $d$ or $\mathbf{B}$, we can write the following free energy functional

$$F = \kappa[(T - T_c + b(\mu_B\mathbf{B})^2)d \cdot d^* + ia\mu_B\mathbf{B} \cdot (d \times d^*) + c\mu_B^2|\mathbf{B} \cdot d|^2 + \alpha(d \cdot d^*)^4 + \eta|d \cdot d|^4] \quad (6)$$

Detailed microscopic derivation of the coefficients $a, b, c, \kappa, \alpha, \eta$ is provided in the Supplementary Note 6. In the absence of spin–orbit coupling, the order parameter's spin is expected to align with the magnetic field. Assuming the magnetic field is

parallel to the $z$-axis, $\mathbf{B} = Be_z$, we can then write

$$d = \left(\frac{\Delta_{\uparrow\uparrow} + \Delta_{\downarrow\downarrow}}{2}, \frac{\Delta_{\uparrow\uparrow} - \Delta_{\downarrow\downarrow}}{2i}, 0\right) \quad (7)$$

Substituting in the free energy (6) and using the fact that $\eta = -\alpha/2$ yields

$$F = \frac{\kappa}{2}\sum_{s=\uparrow,\downarrow} F_s$$
$$F_s = |\Delta_{ss}|^2(T - T_c - \sigma_s a\mu_B B + b(\mu_B B)^2) + \frac{\alpha}{2}|\Delta_{ss}|^4 \quad (8)$$

One important feature is that $\alpha > 0$ which implies the stability of the phase considered.

The free energy (8) leads to the following dependence of the superconducting $T_c$ on the applied field

$$T_{c,\uparrow/\downarrow}(B) = T_c \pm a\mu_B B - b(\mu_B B)^2. \quad (9)$$

The most remarkable feature of this result is that, for nonzero $a$, $T_c$ initially increases upon the application of magnetic field. This can be understood as follows: for a ferromagnetic metal with weakly spin-split Fermi surfaces, the application of the Zeeman field increases (decreases) the density of states for the majority (minority) spin Fermi surface, leading to a linear increase in $T_c$ for the majority spin with the coefficient

$$a = 2\chi T_c \frac{N'(0)}{N(0)} \ln\frac{\Lambda}{T_c} \quad (10)$$

where $\Lambda$ is the bandwidth, $N(0)$ is the density of states at the Fermi energy, and $\chi$ is the dimensionless magnetic susceptibility (Supplementary Note 6). Similar linear field-dependence of $T_c$ is known in superfluid He₃[51], indicating independent pairing for each spin species. This behavior is in stark contrast to the monotonic decrease of $T_c$ under increasing $B$-field in a spin-singlet superconductor. One crucial observation here is that $a$ seems to depend on several details and is expected to be very small since $T_c \ll \frac{N(0)}{N'(0)} \sim \epsilon_F$. Surprisingly, the measured value of $a$ is of order 1[20], which suggests the vicinity of a quantum critical point where the scaling of the susceptibility cancels exactly against the other parameters. Indeed, the scaling $\chi \sim \epsilon_F/(T\log T)$ predicted by Herz-Millis theory in the quantum critical regime

for an itinerant ferromagnet[52,53] leads to such cancellation resulting in $a \sim 1$.

The origin of the quadratic term in Eq. (9) can be understood in terms of the in-plane orbital effect discussed in Sec. IA. First, note that Zeeman splitting cannot break Cooper pairs between aligned spins. Instead, it yields an initial linear increase in $T_c(\mathbf{B})$ followed by saturation at large fields when all the spins are aligned. On the other hand, the in-plane orbital effect can induce pair breaking by mismatching the energies of time-reversal partner states in opposite valleys, resulting in a quadratic decrease in $T_c$ with the applied field whose coefficient is given by (see Supplementary Note 6)

$$b = \frac{1}{T_c} \int_{FS} d\mathbf{k}(e_\mathbf{B} \cdot g_{+,\mathbf{k}})^2 \tag{11}$$

where $e_\mathbf{B}$ is the direction of the external magnetic field. The average value of $(e_\mathbf{B} \cdot g_+(\mathbf{k}))^2$ over the Fermi surface depends strongly on the filling and the field direction with typical value around 1 (cf. Fig. 3d–f). Using this value, we can make a rough estimate for the in-plane field needed to destroy superconductivity as $\mu_B B_c \sim \sqrt{T_c/b}$ yielding a value about 3 Teslas, which compares favorably to the experimental value[20]. Furthermore, if we consider an out-of-plane field instead, $|g_z|$ is on average ~1–2 orders of magnitude larger than $|g_{x,y}|$, yielding a critical field of $\sim 0.1T$ which is very close to the experimentally observed result[20].

It is worth noting that the reduction of $T_c$ at large field can also arise from the suppression of ferromagnetic fluctuations responsible for the pairing, as has been observed in the ferromagnetic superconductor UCoGe[54]. Such effects are neglected within our simplified analysis 3, which assumes a constant coupling $g$.

## Discussion

In this work, we theoretically investigated the physics of twisted double-bilayer graphene (TDBG), addressing the experimental observations of correlated insulating phases at integer fillings and the neighboring superconductor reported in ref. [20].

First, let us summarize a few important features of the band structure. Due to the absence of a $C_2$ symmetry in TDBG, isolated conduction and valence bands with nonzero valley Chern numbers can exist. Moreover, trigonal warping and particle–hole asymmetry in each bilayer graphene lead to (i) a significant broadening of each band so that they overlap in the absence of a displacement field, and (ii) asymmetry between electron- and hole-doped systems. As a result, the parameter space that can host strongly correlated physics is significantly constrained, and the tunability from displacement field at a particular filling becomes essential to realizing correlated states.

Second, we identified an important role played by the coupling of in-plane field to the orbital motion of the electron in TDBG. Despite being small compared with the bandwidth, this effect is comparable with Zeeman splitting, leading to a modified $g$-factor which compares favorably to the experimental value[20] extracted from the slope of the half-filling gap as a function of in-plane field. Moreover, in our theory, this effect is responsible for the reduction of $T_c$ under an in-plane field by providing the main pair-breaking mechanism when pairing takes place between aligned spins in opposite valleys. The resulting decrease in the superconducting $T_c$ with in-plane field agrees qualitatively with the experimental results.

Furthermore, we have performed a self-consistent Hartree–Fock mean-field calculation to identify the possible symmetry broken correlated insulating states at integer fillings. Our prediction of a spin-polarized ferromagnet at half-filling is consistent with the observed increase in the gap with in-plane field.

Finally, here we have proposed a pairing mechanism based on ferromagnetic fluctuations, which is motivated by the evidence for a ferromagnetic parent insulator. Such a mechanism leads naturally to the spin-triplet pairing suggested by experiments. In addition, we showed that the experimentally observed dependence of $T_c$ on in-plane field suggests that the superconductor emerges in the vicinity to a quantum critical point.

In conclusion, our theoretically established phase diagram for twisted double-bilayer graphene, captures all significant observations of the experiments reported in ref. [20]. This includes single-particle features such as the parameter range for band isolation as well as correlation-induced features including a ferromagnetic insulator at half-filling which leads to a spin-triplet superconductor upon doping. In addition to deepening our understanding of correlated Moiré materials, our results highlight how phases which are rare in conventional solids can be readily realized in this novel and tunable platform.

After completing this work, we noticed two experimental papers[55,56] which are consistent with ref. [20] and theoretical discussion contained here.

## Methods

**Numerical simulations for single particle**. Here, we summarize the numerical methods used to calculate the single-particle physics. First, each bilayer-graphene (BLG) layer is modeled by the following bloch Hamiltonian:

$$h_\mathbf{k} = \begin{pmatrix} U_1 + \Delta & -\gamma_0 f(\mathbf{k}) & \gamma_4 f^*(\mathbf{k}) & \gamma_1 \\ -\gamma_0 f^*(\mathbf{k}) & U_1 & \gamma_3 f(\mathbf{k}) & \gamma_4 f^*(\mathbf{k}) \\ \gamma_4 f(\mathbf{k}) & \gamma_3 f^*(\mathbf{k}) & U_2 & -\gamma_0 f(\mathbf{k}) \\ \gamma_1 & \gamma_4 f(\mathbf{k}) & -\gamma_0 f^*(\mathbf{k}) & U_2 + \Delta, \end{pmatrix}, \tag{12}$$

which is labeled in the order of $A_1, B_1, A_2, B_2$. Here, we consider a realistic model of BLG illustrated in Fig. 1. AB stacking means that the $A$-site of the first layer ($A_1$) sits on top of the $B$-site of the second layer ($B_2$). This gives a small on-site energy $\Delta$ for these sites. Here, $f(\mathbf{k}) \equiv \sum_l e^{-i\mathbf{k}\cdot\delta_l}$, where $\delta_1 = a(0, -1)$, $\delta_2 = a(-\sqrt{3}/2, 1/2)$, and $\delta_3 = a(\sqrt{3}/2, 1/2)$ are vectors from $B$-site to $A$-sites. One can expand $f(\mathbf{k})$ near $\mathbf{K}_\pm = \pm(4\pi/3\sqrt{3}a, 0)$ as

$$f(\mathbf{K}_\pm + \mathbf{k}) = \frac{3}{2}(\mp k_x + ik_y)a, \tag{13}$$

where $a$ is the distance between carbon atoms. Throughout, we will use the phenomenological parameters extracted from ref. [57]

$$(\gamma_0, \gamma_1, \gamma_3, \gamma_4, \Delta) = (2610, 361, 283, 138, 15) \text{ meV}, \tag{14}$$

where $\gamma_{0,1,3,4}$ and $\Delta$ are the parameters illustrated in Fig. 1. In addition, the potential difference between the top and bottom graphene layer, $U$ is an important parameter in the experiment, which is controlled by the gate voltage difference. For a displacement field strength $D$, AB–AB system's dielectric constant $\epsilon$ and the thickness of the BLG/BLG system $d$, $U = \epsilon^{-1} D \cdot d$.

Next, we couple two layers of AB-stacked bilayer graphenes by Moire hoping terms. As we are interested in the physics near charge neutrality point, we focus on band structures mostly originated near $\mathbf{K}_\pm$ points. In the continuum model approximation[30], Moire bands from $\mathbf{K}_\pm$ valleys decouple; for the Moire band from $\mathbf{K}_+$ valley, the Hamiltonian is given by

$$H_+ = \sum_\mathbf{k} \left[ h^t_{\frac{\theta}{2}}(\mathbf{K}_+ + \mathbf{k})c^\dagger_{\mathbf{k},+,t}c_{\mathbf{k},+,t} + h^b_{-\frac{\theta}{2}}(\mathbf{K}_+ + \mathbf{k})c^\dagger_{\mathbf{k},+,b}c_{\mathbf{k},+,b} \right.$$
$$\left. + \sum_n \left( T_n c^\dagger_{\mathbf{k}+q_n,+,b}c_{\mathbf{k},+,t} + T_n^\dagger c^\dagger_{\mathbf{k},+,t}c_{\mathbf{k}+q_n,+,b} \right) \right], \tag{15}$$

where $c^\dagger_{\mathbf{k},+,t/b}$ is a 4-components electron creation operator for top/bottom layer with momentum $\mathbf{K}_+ + \mathbf{k}$. Here, $h_\theta(\mathbf{k}) = h(R_\theta\mathbf{k})$ with $R_\theta$ denoting the counter-clockwise rotation matrix by angle $\theta$ relative to the $x$-axis. The momenta $q_{0,1,2}$ are given by $q_0 = R_{\theta/2}K - R_{-\theta/2}K = \frac{8\pi \sin(\theta/2)}{3\sqrt{3}a}(0, -1)$, $q_1 = R_\phi q_0$, and $q_2 = R_{-\phi}q_0$ where $\phi = 2\pi/3$. The hopping matrices $T_n$, $n = 0, 1, 2$ are given by

$$T_n = \begin{pmatrix} 0 & 1 \\ 0 & 0 \end{pmatrix}_{layer} \otimes (w_0 + w_1 e^{2\pi i n\sigma_3/3}\sigma_1 e^{-2\pi i n\sigma_3/3})_{sublattice}, \tag{16}$$

where $w_0, w_1$ are Moiré hopping parameters. One crucial parameter tunable in experiments is displacement field $U$. In Fig. 6, we demonstrated how the band structure evolves with increasing $U$. One can see that the first conduction band becomes isolated in the range of $U \in [40, 80]$. Furthermore, to illustrate the how the band isolation arises, we plot the energy gap between different bands in Fig. 7.

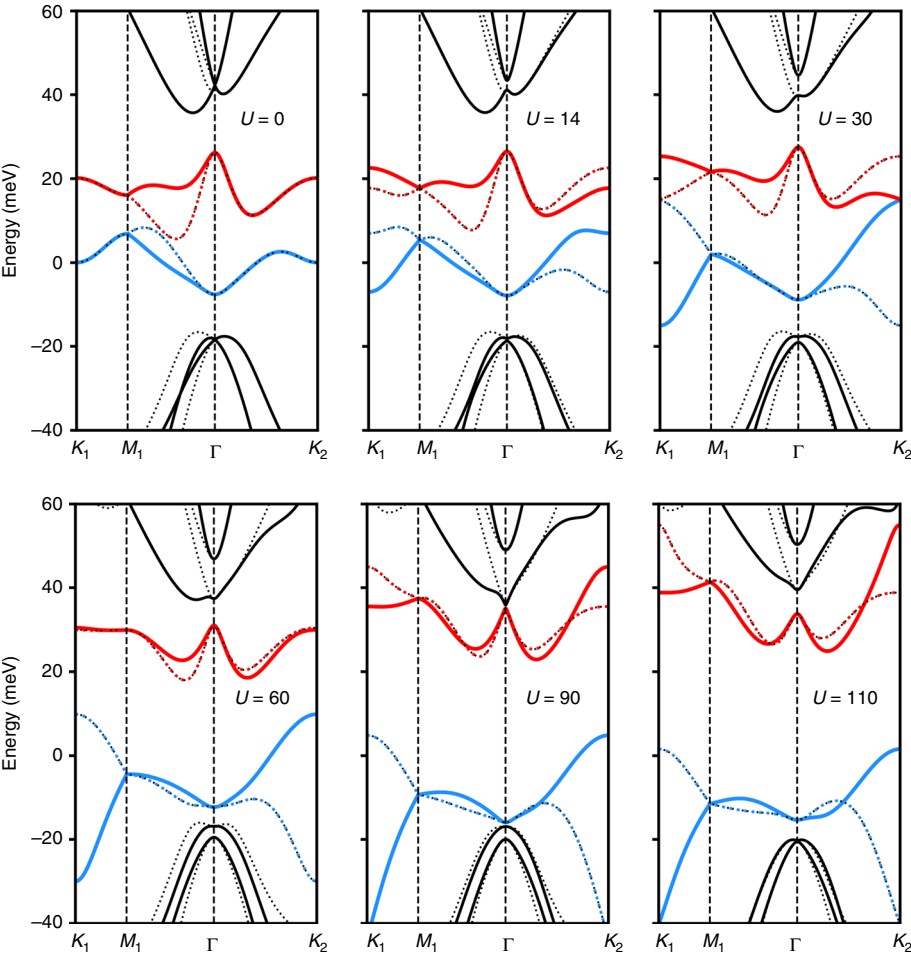

**Fig. 6** The band structure of the model at $\theta = 1.33°$ and $U = 0, 14, 30, 60, 90, 110$. At $U = 14$, Chern number is exchanged by 3 between the conduction and valence band at three momenta which are located not along the symmetric cut. However, at $U = 30$ and $U = 90$, Chern number changes by 1 which can be seen by the gap closing between bands at $K_2$ and $\Gamma$ points.

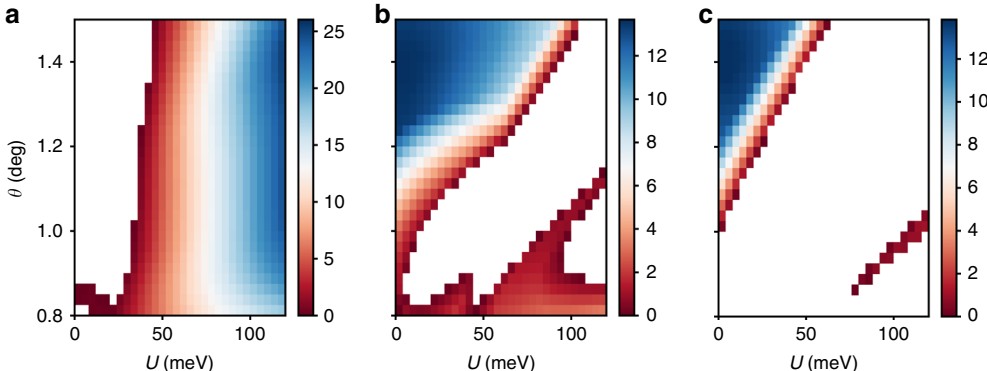

**Fig. 7** The bandgap (meV) for the range of $(\theta, U)$. Uncolored region implies bands being overlapped. **a** Gap between the first conduction and valence bands. **b** Gap between the first and second conduction bands. **c** Gap between the first and second valence bands.

For a smaller value of $r$, gapped regimes in Fig. 7a–c expand in the parameter space of $(\theta, U)$, giving arise to a wider band isolation regime (data available upon request).

**Chern number**. In the main text, we presented Chern number carried by Moire first conduction bands from $K_\pm$-valleys. Here, we carefully examine the evolution of Chern. First, at $U = 0$, the reflection symmetry $M_y$ enforces $C = 0$ for both valleys as $M_y$ maps the system back to itself without exchanging valleys, but $k_y \mapsto -k_y$ so Berry curvature flips its sign[24]. In the quadratic band approximation limit of BLG, as we increase $U$, the band inversion between conduction and valence

bands occurs at the Moiré $K_2$-point ($K_1$ for negative $U$) with a quadratic touching. Thus, Chern number of $\pm 2$ is exchanged.

Next, let us understand the Chern number evolution in the realistic Hamiltonian with parameters of Eq. (14) along the dotted line in Fig. 3b. With a trigonal warping term, the quadratic band touching point splits into four Dirac cones, three with positive and the other with negative chirality. These three Dirac cones are located at generic momenta, thus would not be observed in the band plot along the high symmetric line. Under the presence of particle–hole asymmetry terms, the degeneracy between four Dirac cones split, and the band inversion would happen first at three Dirac cones, exchanging Chern number by $\pm 3$. Then, the band inversion would occur at the center Dirac cone, exchanging Chern number by $\mp 1$. In total, it will still change the Chern number by $\pm 2$. At

larger values of the gate voltage $U$, the band inversion happens between first and second conduction band at $\Gamma$ point, and the Chern number then changes by $\mp 1$ (It can change by $\mp 2$ for other parameter setting), decreasing the Chern number.

This can be further checked by inspecting symmetry indicators[58–60]. There are three $C_3$-invariant momenta $\Gamma$, $K$, and $K'$. For a Bloch state with these momenta, $C_3$ rotation symmetry would map the state back to itself with a rotation eigenvalue:

$$R_{2\pi/3}|\mathbf{k}, n\rangle = e^{2\pi i L_{n,\mathbf{k}}/3}|\mathbf{k}, n\rangle, \quad \mathbf{k} = K_1, K_2, \Gamma \quad (17)$$

where $L_{n,\mathbf{k}}$ is an angular momentum associated with the Bloch state $|\mathbf{k}, n\rangle$. Then, the Chern number of the $n$-th band can be determined modulo 3 by

$$C_n \equiv L_{n,\Gamma} + L_{n,K_1} + L_{n,K_2} \mod 3 \quad (18)$$

Thus, by tracking how $C_3$ eigenvalues of the three momenta change with the gating voltage $U$, we can understand how Chern number transition happens in the system. Indeed, the aforementioned scenario can be confirmed. For example, consider a Moiré first conduction band for $K_+$ valley at $\theta = 1.33°$. At $U = 0$ meV, we start with $(n_\Gamma, n_{K_1}, n_{K_2}) = (0, 1, -1)$. At $U = 14$ meV, Chern number changes by $+3$ but it can be only captured by Berry curvature not by symmetry indicator. At $U = 30$ meV, Chern number changes by $-1$, manifested by $n_{K_2} : -1 \mapsto 1$. At $U = 90$ meV, Chern number again changes by $-1$, manifested by $n_\Gamma : 0 \mapsto -1$. See Fig. 6 for the detail.

**Magnetic field effect.** Under in-plane magnetic field $\mathbf{B} = (B_x, B_y, 0)$, one can choose the gauge $\mathbf{A}(z) = -z \times \mathbf{B}$. Then, the effect of a magnetic field on hopping terms is evaluated via Peierl's substitution, where the hopping term from $R$ to $R + \delta$ is multiplied by the phase factor

$$e^{\frac{iq}{\hbar}\int_R^{R+\delta} dr \cdot \mathbf{A}(z)} = e^{-i\frac{q}{\hbar}\delta_{xy} \cdot \left[\left(R_z + \frac{\delta_z}{2}\right) \times \mathbf{B}\right]}, \quad (19)$$

such that

$$\sum_{R,\delta} e^{\frac{iq}{\hbar}\int_R^{R+\delta} dr \cdot \mathbf{A}(z)} c_{R+\delta}^\dagger c_R = \sum_{\mathbf{k},\delta} e^{-i(\mathbf{k}+\alpha) \cdot \delta} c_{\mathbf{k}}^\dagger c_{\mathbf{k}}, \quad (20)$$

where $\alpha = -\frac{q}{\hbar}\mathbf{A}(R_z + \delta_z/2) = -\frac{e}{\hbar}\left[\left(R_z + \frac{\delta_z}{2}\right) \times \mathbf{B}\right]$ since $\mathbf{A}(z)$ is linear function of $z$. Hence, the effect of in-plane field can be included by simply replacing all $\mathbf{k}$-dependent matrix elements of Bloch Hamiltonians by $\mathbf{k} + \alpha$ as follows (we take $c_{\mathbf{k}} = \sum_R e^{-i\mathbf{k}\cdot R} c_R$):

$$\mathcal{H}_{l,m}(\mathbf{k}, \mathbf{B}) = \mathcal{H}_{l,m}\left(\mathbf{k} - \frac{e}{\hbar}\frac{(l+m)d}{2} e_z \times \mathbf{B}\right) \quad (21)$$

where $\mathcal{H}_{l,m}$ is the matrix element connecting layers $l$ and $m$ ($l, m = 0, \ldots, 3$ from bottom to top) in Eq. (15), $d = 3.42$ A is the interlayer distance, and $e_z$ is the unit vector in the $z$ direction.

Due to its small magnitude relative to the energy gap, it suffices to consider the in-plane orbital effect to first order in perturbation theory. This amounts to adding the following in-plane orbital term to the single-particle energies

$$\xi_{n,\tau}(\mathbf{k}, \mathbf{B}) = \xi_{n,\tau}(\mathbf{k}) + \mu_B g_{n,\tau}^{xy}(\mathbf{k}) \cdot \mathbf{B} \quad (22)$$

where $g_{n,\tau}^{xy}(\mathbf{k})$ is given by

$$g_{n,\tau}^{xy}(\mathbf{k}) = \frac{1}{\mu_B}\langle \psi_{n,\tau}(\mathbf{k})|\nabla_\mathbf{B}\mathcal{H}_\tau(\mathbf{k}, \mathbf{B})|_{\mathbf{B}=0}|\psi_{n,\tau}(\mathbf{k})\rangle, \quad (23)$$

where $\tau$ is the valley index. Time-reversal symmetry implies that $g_{n,\tau}^{xy}(-\mathbf{k}) = -g_{n,-\tau}^{xy}(\mathbf{k})$. The in-plane orbital $g$-factor transforms under $C_3$ rotation as

$$g_{n,\tau}^{xy}(R_{\pm 2\pi/3}\mathbf{k}) = R_{\mp 2\pi/3}g_{n,\tau}^{xy}(\mathbf{k}) \quad (24)$$

provided that the band $n$ is non-degenerate at $\mathbf{k}$. This implies that $g_{n,\tau}^{xy}(\mathbf{k})$ vanishes at any $C_3$-invariant point. As pointed out in the Results, in general, the in-plane orbital contributions affects the bands very differently from the Zeeman effect. For example, it can distort the Fermi surface when the bands are partially filled in an opposite way in the two valleys which can influence the physical properties, e.g., superconducting $T_c$ (see Supplementary Note 6).

The effect of out-of-plane field on the energy bands is generally more complicated since any gauge choice breaks translation symmetry. As a result, the band picture breaks down for large enough out-of-plane fields where Landau level physics form instead. In the following, we will consider the limit of weak out-of-plane fields which can be treated perturbatively. In this case, the out-of-plane field induces an orbital valley Zeeman effect as pointed out in ref. [38,39] whose $g$-factor is given by

$$g_{n,\tau}^z(\mathbf{k}) = -\frac{4m}{\hbar^2}\mathrm{Im}\sum_{l\neq n}\frac{\langle n, \tau|\partial_{k_x}\mathcal{H}_\tau|l, \tau\rangle\langle l, \tau, |\partial_{k_y}\mathcal{H}_\tau|n\rangle}{\epsilon_{n,\tau,\mathbf{k}} - \epsilon_{l,\tau,\mathbf{k}}}. \quad (25)$$

In summary, the single-particle energies has the following dependence on

magnetic field

$$\xi_{n,\sigma,\tau}(\mathbf{k}, \mathbf{B}) = \xi_{n,\tau}(\mathbf{k}) + \mu_B(g\boldsymbol{\sigma} \cdot \mathbf{B} + g_{n,\tau}(\mathbf{k}) \cdot \mathbf{B}), \quad (26)$$

where $\boldsymbol{\sigma}$ is the electron spin operator (which is $\pm 1/2$ for up/down spins) and $\tau = \pm$. The valley orbital $g$-factor is defined as

$$g_{n,\tau}(\mathbf{k}) = (g_{n,\tau}^{xy}(\mathbf{k}), g_{n,\tau}^z(\mathbf{k})) \quad (27)$$

We have also assumed that the spin-quantization axis is parallel to the field.

## Data availability
All relevant data are available from the authors upon reasonable request.

## Code availability
All relevant codes are available from the authors upon reasonable request.

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

## Acknowledgements

We thank Shiang Fang, Yahui Zhang, Yizhuang You, Erez Berg, and Bertrand Halperin for helpful discussion. In particular, we thank Mikito Koshino for clarification on his earlier works on BLG parameters. A.V., J.Y., and E.K. were supported by a Simons Investigator Fellowship. P.K., X.L., and Z.H. acknowledge partial support from the Gordon and Betty Moore Foundation's EPiQS Initiative through Grant GBMF4543 and the DoD Vannevar Bush Faculty Fellowship N00014-18-1-2877.

## Author contributions

J.Y.L., E.K., and A.V. contributed to all aspects of this work. S.L. assisted with the numerical calculation. J.Y.L., E.K., and A.V. wrote the paper in consultation with S.L., X.L., Z.H., and P.K.

## Competing interests

The authors declare no competing interests.
