## [Peer Review File · Nature Communications]

Reviewers' comments:

Reviewer #1 (Remarks to the Author):

The paper "Theory of correlated insulating behaviour and spin-triplet superconductivity in twisted double bilayer graphene" is a systematic study of the twisted double bilayer graphene (TDBG), including its normal phase, correlated insulating phase and also unconventional superconducting phase. For the normal phase, the authors employ the continuum model which is the standard method in moire systems with small lattice mismatch. In the correlated insulating phase and unconventional superconducting phase, the authors mostly apply mean field theory, together with appropriate qualitative arguments when mean field theory is insufficient (for example the unusually strong coupling between magnetic field and Cooper pair spin). The manuscript is well organized and clearly written, the logic is sound, and the results are reasonable and seem to be consistent with experiments. I recommend this paper to be published in Nature Communications. Some minor comments are given as follows.

1. In the normal phase, the authors use continuum model to describe electronic states, and lattice relaxation is incorporated by different hopping integrals in AA and AB regions. However, since in graphene systems the low-energy physics is at K and K' points, lattice relaxations can also introduce pseudo magnetic field. This field may be small near the origin (locally looks like AB stacking and has lower energy), but may be strong near AA regions where the total energy is higher without relaxation. I'm wondering how the relaxation-induced pseudo magnetic field can modify the results, such as band isolation region, valley Chern numbers and magnetic field response.
2. In the correlated insulating phase, the authors seem to assume that translation symmetry is always preserved. Theoretically, what will be the main difference between translation-symmetric insulating phase and translation-breaking insulating phase? Experimentally, besides the evidence mentioned in the manuscript, is there more direct evidence to support translation symmetry?
3. Some minor typos. In Sec. II A, the first sentence "We consider a system consisting of two AB-stacked graphene bilayers twisted relative to ABAB stacking by a small angle θ , illustrated in Fig. 1." In page 7, above the paragraph Chern Number, "For a smaller value of r , gapped regimes in Fig. 7 a,b,c expand in the parameter space of (U, \dots) ".

Reviewer #2 (Remarks to the Author):

In the present manuscript, the authors investigated the electronic properties of twisted double bilayer graphene. They derived the electronic band structure of the physical system. Moreover, using mean-field calculations, they found several novel phases. At half-filling, they found spin polarised and valley polarised phases. Away from half-filling, they found the spin-triplet superconductor phase and obtain the magnetic field dependence of critical temperature. The results, as the authors claimed in the manuscripts, are intriguing.

Before I can come up with my final decision, the authors need to answer the following questions:

(1) To perform the mean-field calculation, they considered the screened Coulomb interaction and claimed that the primary source of screening is from the gate. They need to clarify this claim by providing references for the detail experiment setup. The screened Coulomb potential for A-B stacking bilayer graphene depends mainly on the temperature and the charge density as one can find in M. Lv and S. Wan, Phys. Rev. B 81, 195409 (2010). I expect that the screening Coulomb potential in twisted double bilayer graphene shares similar property.

(2) I think there are some problems with equation (22), in which they used to obtain the extra contribution of the wave function on Lande' g-factor.

- There are two m in this equation. I think one of them is a typo, which should be n, τ . The author should explain the meaning of the remaining m .

- One can derive a very similar equation for the modified g-factor in a semiconductor using k.p perturbation theory (one can look it up in the classic book "Quantum theory of solids" by Charles Kittel). In this derivation, there is no momentum dependence. However, in this paper, they claimed that they found the momentum dependent g-factor. They need to explain in detail the derivation of equation (22).

(3) In the calculation of gap function for singlet and triplet superconductor, the author used "linearized BCS equation" (equation (S72)) which I don't understand where it comes from. If it is the BCS gap equation, then it needs to be a non-linear equation. The linearized BCS equation is only used to derive the critical temperature, in which one can assume the order parameter is small.

(4) I don't understand the Ginzburg-Landau free energy in equation (5). This free energy only has a quadratic term of the order parameter, how can one obtain the superconducting phase transition with this free energy? The author may argue that they only want to derive the critical temperature, so they only need the quadratic term. However, if they wish to claim there is a spin-triplet superconducting phase, they need to calculate higher-order terms.

(5) In the derivation of valley polarized and spin-polarized state, what is the reason that $\kappa_{\mathbf{k}}$ can only equal 1/2?

Further points

- There are several typos in the supplement material; the authors need to check them more carefully.

- The conclusion in the main text is too short, which fails to summarize the main contributions of the paper.

Reviewer #3 (Remarks to the Author):

In this paper, the authors present a comprehensive theoretical analysis of twisted double bilayer graphene by considering low energy effective tight-binding model focusing on the AB and AA sites and the effect of screened Coulomb interaction for the ferromagnetism and effective ferromagnetic spin-spin interaction for superconductivity. As interactions are treated at the mean-field level, obviously the choice of interaction dictates which symmetry breaking can be described. Nevertheless, the authors make a reasonable case for their choice of interactions. Given how any truly microscopic model will be extremely complicated due to a large number of degrees of freedom and the large unit cell, I think the physical intuition driven coarse-grained approach the authors take is quite sensible.

The authors report on three separate aspects of the problem: the band structure, ferromagnetism, and superconductivity, in that order. The key messages are 1) there is no magic angle 2) non-trivial aspects of the band structure requires the application of voltage U which turns the system into a valley-Chern insulator. A hidden message is that the main effect of angle is in tuning the hopping strength, 3) screened Coulomb repulsion can lead to ferromagnetic order at mean-field level, 4) doped ferromagnetic insulator

I am inclined to recommend the paper for publication in Nature Communications after the authors consider the following suggestions for the benefit of the readers.

(1) Although many different order parameters were considered in this paper, given that mean-field theories generally allow for insight, I suspect the inner workings of the competition between different orders can be made more transparent. Although the authors present the detail in the supplementary, once 15 possible order parameters are narrowed to three (SP, VP, and IVC) at the end of p 4, I suggest sharing more explicit insight with readers on why which interaction promotes which ordering rather than simply reporting the results of self-consistent MFT machinery. This can be done for instance by mean-field decoupling the interaction terms in specific channels for instance.

(2) Non-zero valley Chern number in the presence of voltage U has the same physical origin as the AB stacked bilayer in the presence of voltage U that had been actively studied theoretically (Martin et al, PRL 100, 036804 (2008), Vaezi et al PRX 3, 021018 (2013), Zhang et al PNAS 110, 10546 (2013)) and experimentally (Ju et al Nature 520, 650 (2015)) except that the twist angle changes the hopping strengths. It will be helpful to give the readers a larger context, especially many readers who entered the field since the 2018 discoveries may not be aware of previous developments.

(3) The proposed dominance of spin-triplet pairing must be tied to interaction term that is heuristically proposed in Eq (3) that amounts to ferromagnetic fluctuation. I agree with the author it is reasonable to assume the presence of ferromagnetic fluctuation. But once such interaction is assumed, it must be once again possible to trace the observed dominance in pairing spin-triplet pairing to the form of the interaction (and band structure). Although the detail of the calculation is given in the supplement, readers will benefit from gaining insight without having to repeat the calculation. Hence I suggest the authors give insight and rationale one gains upon solving the BCS mean-field theory.

(4) There had been discussions on superconducting order parameters enabled by having valley degree of freedom in transition metal dichalcogenides (TMD) community. I believe the readers will benefit from additional context of TMD superconductivity literature if they could add a few references on TMD superconductivity, although a strong spin-orbit coupling plays a significant role in TMD's.

Response to Referee #1:

1. In the normal phase, the authors use continuum model to describe electronic states, and lattice relaxation is incorporated by different hopping integrals in AA and AB regions. However, since in graphene systems the low-energy physics is at K and K' points, lattice relaxations can also introduce pseudo magnetic field. This field may be small near the origin (locally looks like AB stacking and has lower energy), but may be strong near AA regions where the total energy is higher without relaxation. I'm wondering how the relaxation-induced pseudo magnetic field can modify the results, such as band isolation region, valley Chern numbers and magnetic field response.

As the referee points out, lattice relaxation gives rise to two effects: (i) out-of-plane corrugation changes the ratio of AA/AB hopping terms (ii) in-plane distortion within the Moire unit cell would give a local strain, which gives a local flux pattern. Such a local flux pattern is calculated in the paper **PRB 96, 075311 (2017)**. In most of the literatures, since the net strain (or net flux) within each Moire unit cell is zero, a first approximation is to assume that this effect is small and ignore this for continuum model calculation. To the best of our knowledge, no standard method to incorporate an effective pseudo-flux pattern into a continuum model is known. It would be an interesting direction to understand how such a local strain pattern can be incorporated.

2. In the correlated insulating phase, the authors seem to assume that translation symmetry is always preserved. Theoretically, what will be the main difference between translation-symmetric insulating phase and translation-breaking insulating phase? Experimentally, besides the evidence mentioned in the manuscript, is there more direct evidence to support translation symmetry?

In principle, as the referee suggests, translation symmetry can be spontaneously broken, for example forming a charge density wave (CDW) order. However, if half-filling insulator appears due to the translation symmetry breaking (by enlarging a unit cell twice), it does not need to break other symmetries (valley and spin). Thus, a paramagnetic phase may be expected, where the gap is supposed to decrease upon applying a Zeeman field. However, the gap increased as a function of Zeeman field, which supports the ferromagnetic ordering over other symmetry broken phases. More direct evidence would be the STM image, which is currently unavailable for TDBG, but in TBG, the image seems to be translational symmetric from one Moire unit cell to the next, despite breaking symmetries within the unit cell.

3. Some minor typos. In Sec. II A, the first sentence “We consider a system consisting of two AB-stacked graphene bilayers twisted relative to ABAB stacking by a small angle θ , illustrated in Fig. 1.” In page 7, above the paragraph Chern Number, “For a smaller value of r , gapped regimes in Fig. 7 a,b,c expand in the parameter space of (U, \dots) ”.

We really appreciate the effort to point out our typos. In the revised version, all typos are corrected to the best of our knowledge.

Reviewer #2 (Remarks to the Author):

(1) To perform the mean-field calculation, they considered the screened Coulomb interaction and claimed that the primary source of screening is from the gate. They need to clarify this claim by providing references for the detail experiment setup. The screened Coulomb potential for A-B stacking bilayer graphene depends mainly on the temperature and the charge density as one can find in M. Lv and S. Wan, Phys. Rev. B 81, 195409 (2010). I expect that the screening Coulomb potential in twisted double bilayer graphene shares similar property.

Thanks for pointing this out. In the revised manuscript, the reference for a screened Coulomb potential is correctly included. It is correct that the Coulomb potential would also depend on the temperature and charge carrier density; however, here we considered a zero temperature

ground state of the mean-field solution. For the mean-field calculation, we made several approximations to get a qualitatively correct answers; we assumed that the dielectric constant is constant for different momenta and different fillings (quarter and half). Thus, effectively, the change in screening effect due to other charge carrier can be incorporated into different values of dielectric constant. We confirmed that our result is robust under the range of dielectric constants. Solving the problem with more careful treatment (k-dependent dielectric constant) would be beyond the scope of the present work.

(2) I think there are some problems with equation (22), in which they used to obtain the extra contribution of the wave function on Lande' g-factor.

- There are two m in this equation. I think one of them is a typo, which should be n, τ . The author should explain the meaning of the remaining m .

We appreciate the comment. The typo is fixed in the revised version.

- One can derive a very similar equation for the modified g-factor in a semiconductor using k.p perturbation theory (one can look it up in the classic book "Quantum theory of solids" by Charles Kittel). In this derivation, there is no momentum dependence. However, in this paper, they claimed that they found the momentum dependent g-factor. They need to explain in detail the derivation of equation (22).

We believe that the modified g-factor formula the referee mentions semiconductors is from the following reference: **Roth et al., Phys. Rev. 114, 90**. In that reference, the modified g-factor is calculated for the zone center, and there indeed is a momentum dependence. Also, the formula is meaningful only for the materials with large spin-orbit coupling, while the graphene has a negligible spin-orbit coupling. Thus, these contributions to the effect are not important; rather, the additional modification which comes from the orbital effects is much larger. The derivation of Equation (22), as we referred to [34,35], is quite standard and is detailed in the references we cited.

(3) In the calculation of gap function for singlet and triplet superconductor, the author used "linearized BCS equation" (equation (S72)) which I don't understand where it comes from. If it is the BCS gap equation, then it needs to be a non-linear equation. The linearized BCS equation is only used to derive the critical temperature, in which one can assume the order parameter is small.

The referee is correct. The linearized gap equation is used to derive the critical temperature. When there are several possible pairing channels, the linearized BCS equation is used to compute T_c for each channel. We can then identify the dominant pairing channel as the one with the largest T_c , which is a relatively standard analysis (see for example "Introduction to

unconventional superconductivity” by Manfred Sigrist). In fact, in our case, momentum-independent pairing is only possible in the spin-triplet channel. This can be seen by decoupling the interaction in the different pairing channels and showing that it is attractive in the spin-triplet channel but repulsive in the spin-singlet channel. We have added a discussion clarifying this aspect to the main text.

(4) I don't understand the Ginzburg-Landau free energy in equation (5). This free energy only has a quadratic term of the order parameter, how can one obtain the superconducting phase transition with this free energy? The author may argue that they only want to derive the critical temperature, so they only need the quadratic term. However, if they wish to claim there is a spin-triplet superconducting phase, they need to calculate higher-order terms.

As the referee mentions, we only considered the quadratic term to determine the dependence of the critical temperature on applied field. We agree that higher order (quartic) terms are needed to establish the existence of the phase. In the updated manuscript, we have included such quartic terms and showed by explicit computation (in supplemental material) that the quartic term is positive, thus the spin-triplet superconducting phase is indeed a stable phase. We have also expanded the discussion of the Ginzburg-Landau functional to make this part more transparent. We thank the referee for this very useful suggestion.

(5) In the derivation of valley polarized and spin-polarized state, what is the reason that $\kappa_{\mathbf{k}}$ can only equal 1/2?

The reason is that for a fully gapped solution, the trace of the order parameter at every \mathbf{k} has to equal the filling i.e. $\text{tr} M(\mathbf{k}) = \nu$ for all \mathbf{k} (this follows from Eq. S39). We have expanded the discussion of the mean field solutions both in the main text and the supplemental material to clarify this and other related issues.

Further points

- There are several typos in the supplement material; the authors need to check them more carefully.

We appreciate the comment. The typo is fixed in the revised version.

- The conclusion in the main text is too short, which fails to summarize the main contributions of the paper.

We added more detailed summary in the conclusion section.

Reviewer #3 (Remarks to the Author):

In this paper, the authors present a comprehensive theoretical analysis of twisted double bilayer graphene by considering low energy effective tight-binding model focusing on the AB and AA sites and the effect of screened Coulomb interaction for the ferromagnetism and effective ferromagnetic spin-spin interaction for superconductivity. As interactions are treated at the mean-field level, obviously the choice of interaction dictates which symmetry breaking can be described. Nevertheless, the authors make a reasonable case for their choice of interactions. Given how any truly microscopic model will be extremely complicated due to a large number of degrees of freedom and the large unit cell, I think the physical intuition driven coarse-grained approach the authors take is quite sensible.

The authors report on three separate aspects of the problem: the band structure, ferromagnetism, and superconductivity, in that order. The key messages are 1) there is no magic angle 2) non-trivial aspects of the band structure requires the application of voltage $\$U\$$ which turns the system into a valley-Chern insulator. A hidden message is that the main effect of angle is in tuning the hopping strength, 3) screened Coulomb repulsion can lead to ferromagnetic order at mean-field level, 4) doped ferromagnetic insulator

I am inclined to recommend the paper for publication in Nature Communications after the authors consider the following suggestions for the benefit of the readers.

(1) Although many different order parameters were considered in this paper, given that mean-field theories generally allow for insight, I suspect the inner workings of the competition between different orders can be made more transparent. Although the authors present the detail in the supplementary, once 15 possible order parameters are narrowed to three (SP, VP, and IVC) at the end of p 4, I suggest sharing more explicit insight with readers on why which interaction promotes which ordering rather than simply reporting the results of self-consistent MFT machinery. This can be done for instance by mean-field decoupling the interaction terms in specific channels for instance.

We thank the referee for this suggestion. However, due to the complicated form of the interaction and its non-trivial momentum dependence through the form factors, we found that attempting a mean-field decoupling in different channels is not the best way to present an intuitive picture for the competition between different orders. Instead, we expanded the discussion in the main text and the supplemental material to clarify the different possible symmetry breaking orders. Based on the possible symmetries that can be broken, we narrowed down all possible self-consistent solutions to 5 possibilities at half-filling and 3 possibilities at quarter-filling. Using independent spin-rotations in each valley (which is an approximate symmetry that is only violated by the small Hund's term), we can show that only 3 (SP, VP, and OVC) of these 5 possibilities at half-filling and only 2 (SP+VP and SP+IVC) out of the 3 possibilities at quarter filling are distinct. The energy competition between the valley diagonal orders (SP,VP) and valley off-diagonal orders (IVC) can be understood using an analytical

argument adopted from 1901.08110 where particle-hole transformation in only one of the valleys is used to transform the IVC order into a superconductor. After the transformation, the Chern number is the same in both valley and a non-zero Chern number implies the existence of vortices in the superconducting order parameter which are energetically costly and make the IVC state energetically less favorable. We believe the current discussion makes the underlying mechanism behind the different orders and the competition between them more transparent.

(2) Non-zero valley Chern number in the presence of voltage U has the same physical origin as the AB stacked bilayer in the presence of voltage U that had been actively studied theoretically (Martin et al, PRL 100, 036804 (2008), Vaezi et al PRX 3, 021018 (2013), Zhang et al PNAS 110, 10546 (2013)) and experimentally (Ju et al Nature 520, 650 (2015)) except that the twist angle changes the hopping strengths. It will be helpful to give the readers a larger context, especially many readers who entered the field since the 2018 discoveries may not be aware of previous developments.

We appreciate your suggestions. As you correctly pointed out, non-zero valley Chern number comes from the fact that bilayer graphene gapped due to electric field has oppositely accumulate Berry curvatures near K and K' points, and the role of Moire-structure is to decouple them. In the single-particle physics section, we added a proper explanation and suggested references for it.

(3) The proposed dominance of spin-triplet pairing must be tied to interaction term that is heuristically proposed in Eq (3) that amounts to ferromagnetic fluctuation. I agree with the author it is reasonable to assume the presence of ferromagnetic fluctuation. But once such interaction is assumed, it must be once again possible to trace the observed dominance in pairing spin-triplet pairing to the form of the interaction (and band structure). Although the detail of the calculation is given in the supplement, readers will benefit from gaining insight without having to repeat the calculation. Hence I suggest the authors give insight and rationale one gains upon solving the BCS mean-field theory.

We thank the referee for this very useful suggestion. In fact, the dominance of spin-triplet pairing for the proposed interaction can be seen by decoupling the interaction in the different (momentum-independent) pairing channels and finding that it is effectively attractive in the spin-triplet channel and repulsive in the spin-singlet channel. We have added such discussion to the main text.

(4) There had been discussions on superconducting order parameters enabled by having valley degree of freedom in transition metal dichalcogenides (TMD) community. I believe the readers will benefit from additional context of TMD superconductivity literature if they could add a few references on TMD superconductivity, although a strong spin-orbit coupling plays a significant role in TMD's.

We appreciate your suggestions. Indeed, the pairing in TDBG and the pairing in TMD are similar in that electrons from opposite valleys are paired, where the time-reversal symmetry guarantees the perfect nesting between electrons. In the superconductivity section (page 6, first paragraph) of the revised manuscript, we added some discussions on the similarity and difference between TDBG and TMD.

REVIEWERS' COMMENTS:

Reviewer #1 (Remarks to the Author):

I am satisfied with the answers to my questions. However, I would like to raise one question on quartic terms of Eq. (6). From the Supplementary, in Eq. (6) the ratio between coefficients of two quartic terms is -2, independent of the interaction details. I think it may be an artifact due to the single interaction term in Eq (3), since symmetry does not fix the ratio between these two coefficients. Could the authors comment on this?

Besides, there are typos in quartic terms of Eq. (6), the exponent should be 2 instead of 4.

Reviewer #2 (Remarks to the Author):

The authors adequately addressed issues raised in the Referee report and answered referees' question satisfactorily. They also provided more detail discussion and calculation for the superconducting phase.

I have no more questions and recommend to publish the revised version of the manuscript.

Reviewer #3 (Remarks to the Author):

The authors have addressed my concerns and suggestions in a satisfactory manner. I recommend the paper for publication in Nature Communications.

Response to Referee #1:

I am satisfied with the answers to my questions. However, I would like to raise one question on quartic terms of Eq. (6). From the Supplementary, in Eq. (6) the ratio between coefficients of two quartic terms is -2, independent of the interaction details. I think it may be an artifact due to the single interaction term in Eq (3), since symmetry does not fix the ratio between these two coefficients. Could the authors comment on this? Besides, there are typos in quartic terms of Eq. (6), the exponent should be 2 instead of 4.

It is true that the ratio between the coefficients of the two quartic terms is not fixed by symmetry. Instead, it is fixed by the projection onto the spin-triplet channel which yields the form of the Fermionic determinant given in S83. This means that it is independent of the details of the interaction provided the dominant Cooper pairing channel of this interaction is the spin-triplet channel and as long as we neglect the other channels. We also thank the referee for pointing out the typo in Eq. (6). It is indeed correct that the exponent should be 2 instead of 4.

Response to Referee #2:

The referee recommended publications without further comments.

Response to Referee #3:

The referee recommended publications without further comments.